

# Parametric Sensitivity and Constraint of Contrail Cirrus Radiative Forcing in the Atmospheric Component of CNRM-CM6-1

Maxime Perini[1], Laurent Terray[1], Daniel Cariolle[1], Saloua Peatier[1], and Marie-Pierre Moire[1]

[1]CERFACS, Toulouse, France

**Correspondence:** Maxime Perini (maxime.perini@cerfacs.fr)

**Abstract.**

The impact of aviation on climate change due to $CO_2$ emissions no longer needs to be demonstrated. However, the impact of non-$CO_2$ effects such as those from contrails is still subject to large uncertainties. An often neglected source of uncertainty comes from climate model sensitivity to numerical parameters representing subgrid-scale processes. Here we investigate the

sensitivity of contrail radiative forcing due parametric uncertainty based on the atmospheric component of the CNRM-CM6-1 coupled model. A perturbed parameter ensemble is generated from the sampling of twenty-two adjustable parameters involved in convection, cloud microphysics and radiative transfer processes. A surrogate model based on multi-linear regression is used to explore the full range of contrail radiative forcing due to parametric uncertainty. Based on an optimization algorithm and a climatological skill score, we find a constrained range of contrail radiative forcing from equally skillful model versions with

different sets of parameters. We find a contrail radiative forcing best-estimate of 56 $mW.m^{-2}$ with a 5-95 % confidence interval of 38-70 $mW.m^{-2}$. Finally, a sensitivity analysis shows that model parameters controlling contrail's lifetime play a major role in the estimation of contrail radiative forcing.

## 1   Introduction

Contrails induce positive longwave (LW) and negative shortwave (SW) instantaneous radiative forcing components at the

top of the atmosphere (TOA) (Meerkötter et al., 1999), leading to a positive net radiative effect and a warming of the Earth–atmosphere system. The contrail radiative forcing best-estimate for the year 2018 is 111 $mW.m^{-2}$ with a 5-95 % confidence interval of 33 - 189 $mW.m^{-2}$ (Lee et al., 2021). This wide range of values suggests that the complexity of contrail physics is still not well understood leading to large uncertainty estimates. Note that this estimate includes persistent linear contrails and contrail cirrus. However, radiative forcing due to contrail cirrus is the largest component. For example, the study

by Boucher et al. (2013) assessed the radiative forcing from persistent linear contrails and contrail cirrus in year 2011 to be 10 $mW.m^{-2}$ and 40 $mW.m^{-2}$, respectively. Improving the quantification of these uncertainties in order to include non-$CO_2$ effects in global climate policies remains a priority. A substantial fraction of these uncertainties comes from parametric uncertainty, meaning the sensitivity of simulated contrail radiative forcing to adjustable parameters that are used in the model physical parameterizations to represent subgrid-scale processes. Model calibration consists in assigning specific values to these

adjustable parameters and is often based on expert knowledge but remains largely empirical. Another, more objective and more





computing-intensive, way to calibrate a climate model is to generate a perturbed parameter ensemble (PPE), in which model parameters are varied within expert-defined ranges (Bellprat et al., 2012; Hauser et al., 2012). PPEs have been shown to allow a thorough exploration of model parametric uncertainty regarding global climate sensitivity (Piani et al., 2005; Sanderson et al., 2008; Hourdin et al., 2023). Here our main focus is to explore and quantify atmospheric model parametric uncertainty in the estimation of contrail radiative forcing, the model being ARPEGE-climat, the atmospheric component of the CNRM-CM6-1 model (Voldoire et al., 2019). The ARPEGE-climat model includes a parameterization for ice supersaturation and contrails, which initially underestimated by one order of magnitude the contrail radiative forcing (not shown).

Our first objective is to present the original ice-supersaturation/contrails parameterization, based on results from mesoscale simulations and calibrated according to expert judgment. We also detail how we have modified it to reach a better agreement with results from previous studies. Evaluation of the model parameterization performance against different observational datasets is presented.

Our second objective is to explore the parametric dependence of contrail radiative forcing to different calibrations of ARPEGE-Climat. We use the approach developed by Peatier et al. (2022) with a few minor changes. First, a PPE is generated and evaluated by comparing model climatology with observational/reanalysis data. Then, to more densely sample the parameter space at lower computational cost, statistical emulators based on multilinear regressions are used to predict the contrail radiative forcings and the associated model performances. The optimization method aims at identifying model calibrations minimizing the multi-variate model error, while exploring all the contrail radiative forcing range. Finally, these calibrations are run in the ARPEGE-Climat model, their outputs gives an estimate of the parametric uncertainty associated with contrail radiative forcing.

## 2   Model and Baseline Experiment

The PPE is based on ARPEGE-climat version 6.3 (Roehrig et al., 2020), the atmospheric component of CNRM-CM6-1 (Voldoire et al., 2019). ARPEGE-Climat 6.3 (ARPEGE thereafter) used in this work has a spatial resolution of about 150 km, with 91 vertical levels from near 6 m to 82 km (or 0.01 hPa). The ARPEGE released version will be referred to as the ARPEGE baseline version. Alternative versions (meaning with different sets of adjustable parameters) of this model will be called ARPEGE candidates. Experiments under investigation in this work are AMIP-type simulations, where the atmospheric model is forced by observed sea surface temperatures and sea ice concentrations. The emission data used to represent aviation's climate forcing were generated during the project QUANTIFY, an integrated project funded through the EU-Framework Program 6 (https://www.ip-quantify.eu). These data contain monthly totals of fuel consumption and emissions for NOx and soot in $kg.gridbox^{-1}.month^{-1}$ with a horizontal resolution of $1° \times 1°$ and 610 m vertical spacing of up to 14 km. Inventories and scenarios are available for the years 2000, 2025, and 2050.

This work only focuses on the present-day period based on three-year (2000-2002) simulations forced by the aviation emission inventory for the year 2000. The values of twenty-two ARPEGE numerical parameters are varied within expert-defined ranges (Table S1 in Supporting Information) without modifying the calibration of the contrail cirrus parameterization.





The parameter values simultaneously vary and are sampled with a Latin Hypercube to have a uniform sampling of the parameter
space (Urban and Fricker, 2010). Two hundred simulations are run to approximately get ten realizations per parameter as
discussed in Loeppky et al. (2009). A total of 190 experiments have been successfully retrieved from this PPE and are used in
the present study.

## 2.1   Ice supersaturation and contrail cirrus parameterization

In the standard version of ARPEGE, a cloud-free and ice-supersaturated fraction of a grid mesh is not permitted. To include
this possibility, the cloud scheme (Ricard and Royer, 1993) has been modified. The modification is also able to account for the
additional contrail coverage. The standard cloud scheme (the existing scheme thereafter) is based on a Gaussian distribution
of the sub-grid scale humidity to predict the cloud fraction of the grid mesh. The left part of Fig. 1 represents how the existing
scheme works. No supersaturation is allowed, the cloud cover, $a^*$, is given by the integral of the distribution function for
$q > q_{sat}$ (yellow area). In the new scheme (A to F scenario on Fig. 1), a prognostic variable, $x$, is introduced and indicates
the presence or not of ice nuclei in the grid mesh. Then, if there is no ice nuclei ($x = 0$) there is no additional cloud cover if
there is no local supersaturation: $q < k.q_s$ with $k$ indicating the value of the *strong* supersaturation to create a cloud without
preexisting ice nuclei. Initially, $k$ was an adjustable parameter of the method but it is now a function of the temperature
following the empirical approximation from  Koop et al. (2000). For example, in case B (weak supersaturation and no ice
nuclei, $x = 0$), the new cloud scheme cloud cover, $a$, is zero (whereas in the existing cloud scheme, $a^*$, is positive). In this
case, the contrail cirrus parameterization is activated whereas in every other case, the cloud scheme works like the existing one.
Note that the parameterization is only activated for temperature below $233K$  (Schumann, 1996).

The contrail cirrus parameterization is based on previous mesoscale simulations results (Paoli et al., 2017). The fraction of
the grid mesh covered by contrails, $a$, is computed according to the following equation :

$$\frac{da}{dt} = N \frac{dl_t}{dt} \frac{\min(v\delta t, \delta x)}{(\delta x)^2} P \tag{1}$$

with :

1. $P$, the probability for the grid mesh to be saturated = surface of the grid mesh that is saturated in the atmospheric model.

2. $l_t$, the contrail width ($\frac{dl_t}{dt} = 1.5 m/s$)

3. $N$, the number of aircrafts in the grid mesh

4. $\delta x$, the spatial resolution of the model, i.e grid mesh size ($\sim 150 km$)

5. $v$, the mean plane's velocity ($900 km/h$)

6. $\delta t$, the atmospheric model time step ($900 s$)



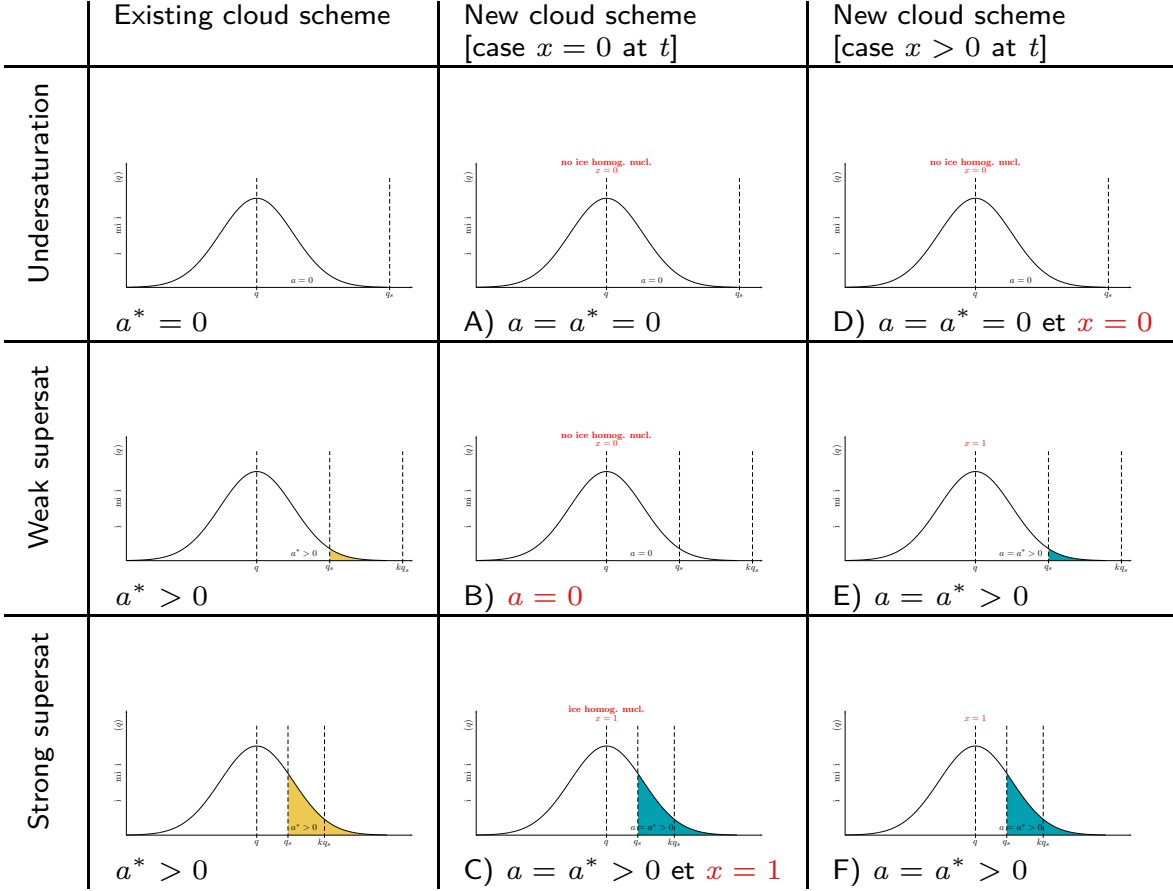

**Figure 1. Schematic of the cloud fraction computation in (left) the old and (center and right) the new cloud scheme in ARPEGE (David Saint-Martin, personal communication). The black curve represents the probability distribution function of the sub-grid specific humidity. The yellow and blue areas represent the cloud fraction computed in the old and new cloud scheme, respectively.**

## 2.2 Model evaluation: the baseline experiment

After introducing the cloud scheme modification and the additional contrail cirrus parameterization, we want to assess the model representation of the distribution and amount of high-level cloud climatology. Therefore, we compare the baseline experiment (that includes the new cloud scheme with the contrail parameterization) and the corresponding simulation performed with the standard (existing) cloud scheme with observations from CALIPSO (Chepfer et al., 2010). We use the CFMIP Observation Simulator Package (COSP) satellite simulator (Bodas-Salcedo et al., 2011) to compare the simulated cloud cover above 440 hPa to CALIPSO VFM (Vertical Feature Mask Version, Vaughan et al. (2009)) data, known for its ability to detect very thin high clouds. Figure 2 shows that in the region of interest, between 60S-60N, where almost all the air traffic takes





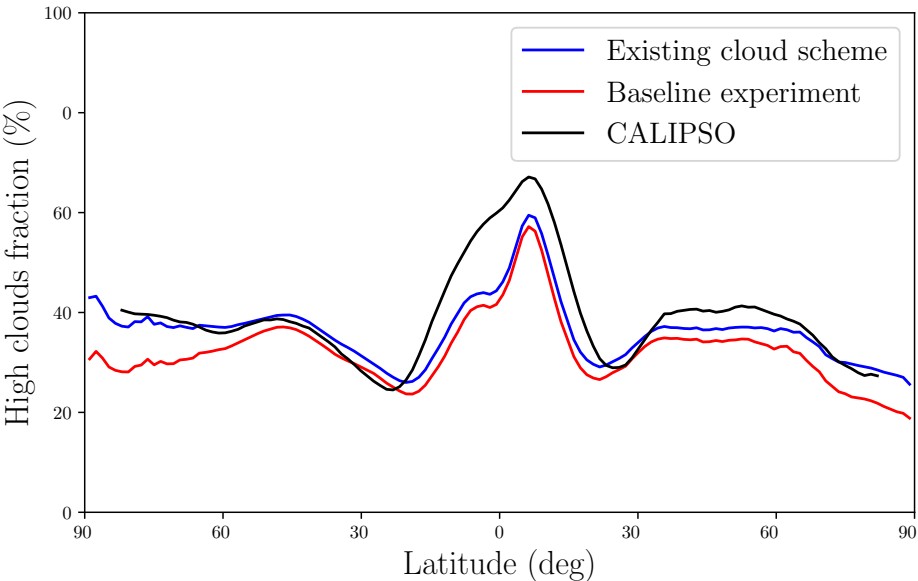

**Figure 2. Annual and zonal mean of high clouds fraction (440 hPa) : CALIPSO data in black, existing cloud scheme in blue and baseline experiment in red.**

place, both cloud schemes give similar results. However, the baseline experiment underestimates the high cloud fraction in polar regions. This can be explained, at least over the North Pole, by the overestimation of the ice-supersaturation frequency over this region. Allowing more supersaturation over ice automatically decreases cloud cover. The same mechanism probably helps to reduce the positive bias of total cloud cover over ice-covered regions (not shown here) possibly related to the missing supersaturation process, as discussed in Roehrig et al. (2020). Figure 3 shows the frequency of ice supersaturation in the upper troposphere in the baseline experiment and in the AIRS (Atmospheric Infrared Sounder) satellite observations from Lamquin et al. (2012). The simulated spatial distribution and amplitude of supersaturation frequency are in reasonable agreement with their observed counterpart. Ice supersaturation frequency follows the tropopause and local maxima are related to the storm tracks and jet streams in the midlatitudes at 200–250 hPa as observed in Lamquin et al. (2012) and Bock and Burkhardt (2016). The ARPEGE model overestimates the supersaturation frequency in the tropics but AIRS measurements over such cloudy regions may also lead to underestimation. The supersaturation frequency is overestimated over the North Pole indicating a too high tropopause compared with the AIRS data. In addition, the new cloud scheme struggles to represent supersaturation over the Antarctic, except over the western part. Again, measurements over icy regions are subject to large uncertainties.

The ARPEGE model is also evaluated on specific regions of interest. Figure 4 shows the PDF of the relative humidity (with respect to ice) for the North Atlantic flight corridor for the MOZAIC observations (Marenco et al., 1998) (available on IAGOS





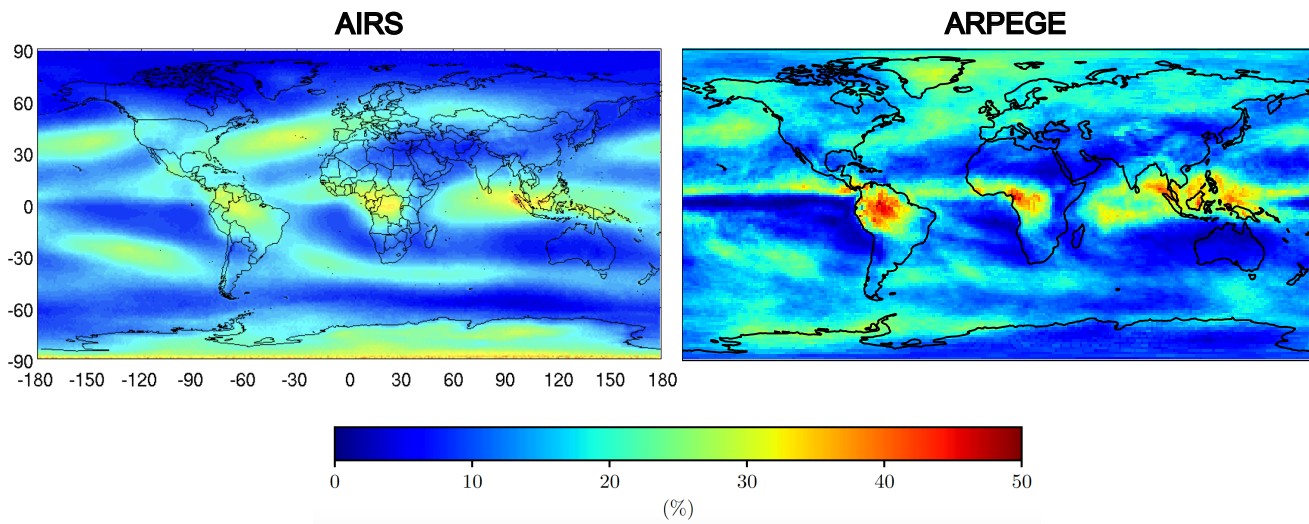

**Figure 3.** Annual mean of frequency of ice supersaturation between 200-250 hPa in (left) scaled AIRS satellite measurements (Lamquin et al., 2012) and (right) in ARPEGE-Climat.

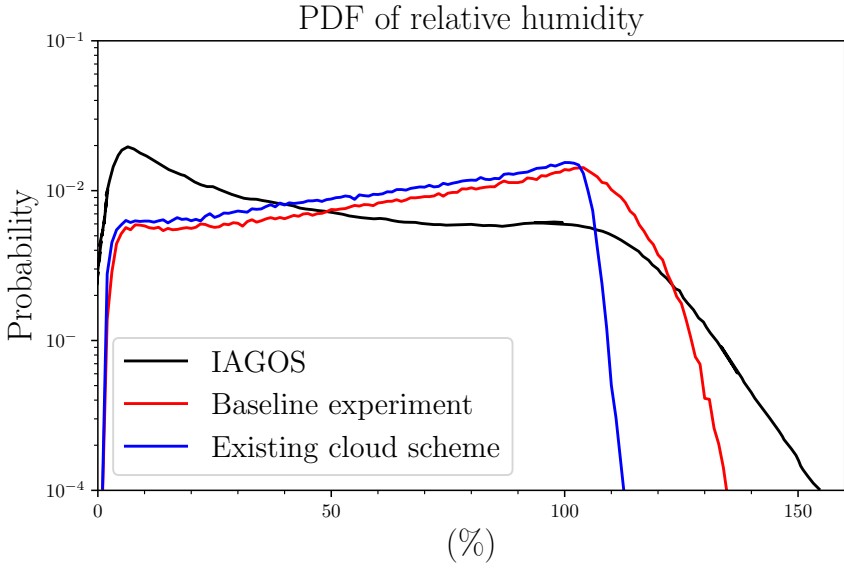

**Figure 4.** Average probability density function of relative humidity with respect to ice for the North Atlantic flight corridor ($40°$N to $60°$N and $65°$W to $5°$W) for the years 2000 to 2009: MOZAIC data in black, existing cloud scheme in blue and baseline experiment in red.





data portal at https://iagos.aeris-data.fr/), the baseline and the old cloud scheme experiments. The data set used for this analysis is constrained to a pressure below 350 hPa and to ambient temperatures below 233K to exclude potential sensor contamination by supercooled liquid water droplets (Petzold et al., 2020). The mode at approximately $10\%$, corresponding to dry stratospheric air mass, is slightly more noticeable in the baseline experiment. After the $100\%$ mode, the baseline experiment shows a better

agreement with the shape of MOZAIC data despite a steeper downward slope.

Finally, we assess the representation of contrails based on annual mean coverage from the year 2000 air traffic data (Fig. 5(a) ). As expected, the denser the traffic, the higher the contrails coverage. The maximum of contrails coverage is observed over Western Europe, the United States, and the North Atlantic. The annual mean is 0.6 % and is in reasonably good agreement with the value of 0.74 % found by Bock and Burkhardt (2016) for 2002 air traffic. However, this value is much higher than 0.11 %

from Rap et al. (2010), for 2002 air traffic.

Figure 5(b) represents the net radiative forcing at the top of the atmosphere due to contrails for the year 2000 air traffic. The long wave and the short wave components are represented in Fig.S1 and Fig.S2 in the Supporting Information. The annual mean of 57.5 $mW.m^{-2}$, only including contrail cirrus, is in good agreement with the last estimate of 67.5 $mW.m^{-2}$ given by Lee et al. (2021), for the year 2005. It is important to keep in mind the large range of the 5-95 % confidence interval of

20-115 $mW.m^{-2}$ which rules out the estimate of 12.9 $mW.m^{-2}$ for the year 2002, from Rap et al. (2010), though it is the same order of magnitude.

To summarize, the modified ice-supersaturation/contrail cirrus parameterization in ARPEGE improves the comparison with observations and leads to an estimate of top of atmosphere contrail radiative forcing in line with the most recent estimate from the literature. The next step is to explore and quantify the influence of atmospheric parametric uncertainty on the estimation of

contrail radiative forcing. In the next section, we use the perturbed parameter ensemble and an optimization algorithm based on a surrogate model of the PPE to quantify the uncertainty range of contrail radiative forcing.

## 3   Perturbed Parameter Ensemble

### 3.1   Performance assessment of model predictions

The model performance is evaluated by comparison between the annual mean climatology from experiments and observation-

al/reanalysis data. Table 1 gives the physical variables and observed datasets used to assess the model performance. Most of the fuel consumption takes place at flight altitude (9-12km) and mid-latitude in the northern hemisphere. Therefore we focus on the Northern Hemisphere upper troposphere mid-latitudes ($30^oN - 60^oN$) to assess the ARPEGE model performance regarding the representation of air temperature.

For each variable, the empirical orthogonal functions (EOFs) are calculated to determine the dominant modes of variability

within the PPE. Contrary to the conventional EOF method, the temporal dimension is replaced by the ensemble dimension itself (Sanderson et al., 2008; Sexton et al., 2012). The resulting EOFs are spatial patterns, characterizing the variability of the ensemble variance, while their principal components (PCs) are the expansion coefficients showing the projection of each ensemble member onto the respective EOF. Then, the annual mean climatology of the observations of a given variable is



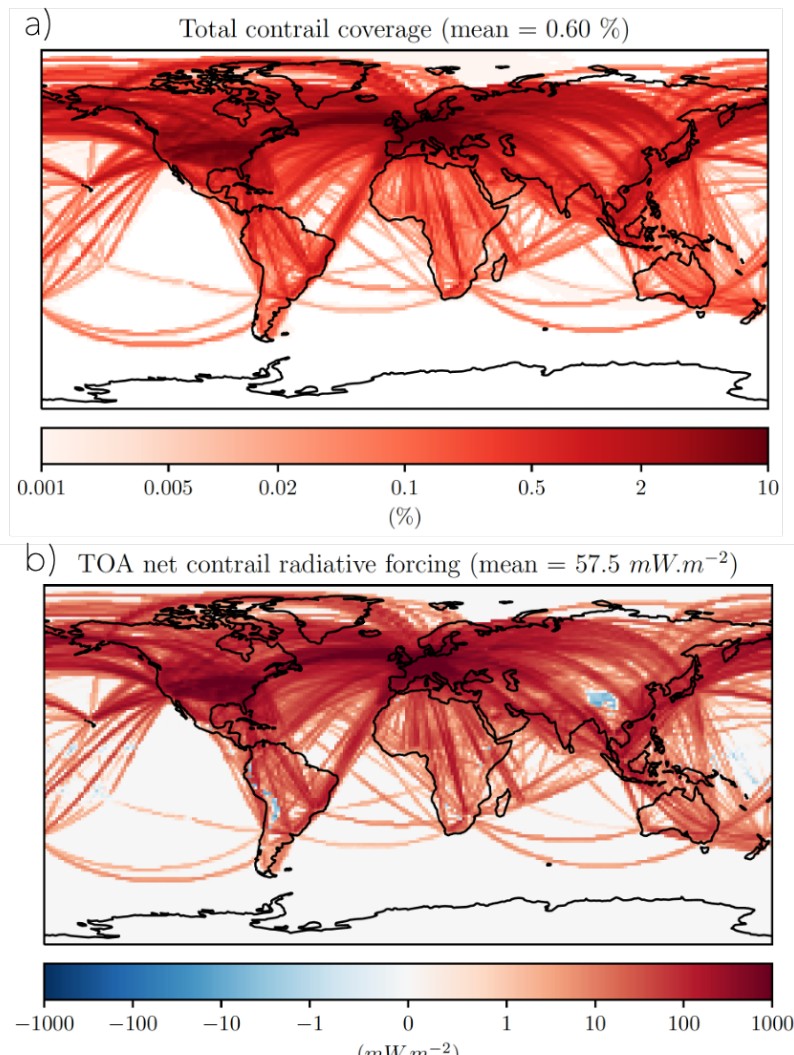

**Figure 5. Annual mean of global contrail coverage and contrail radiative forcing at the top of atmosphere, for the 2000 air traffic.**

projected on the EOF basis and the three-dimensional fields of both model members and observations are reconstructed, using
only the first five modes of the EOFs since it allows to account for 62 %, 92 % and 95 %, for precipitation, radiative fluxes and
temperature respectively, of the PPE variance.

The error score is then represented by the area-weighted RMSE of these reconstructed fields. The error associated with each
variable $E_s$ can be normalized by the error of the baseline simulation $E_{ESM,s}$, allowing the combination of errors associated
with the different variables and the estimation of a total error $E_{tot}$:



**Table 1.** Observational datasets used for performance assessment

| Variable | Units | Dataset | Reference | Time period |
|---|---|---|---|---|
| rsut - SW radiation at TOA | $W.m^2$ | CERES edition 4.1 | (Loeb et al., 2018) | 2000-2019 |
| rlut - LW radiation at TOA | $W.m^2$ | CERES edition 4.1 | (Loeb et al., 2018) | 2000-2019 |
| ta - Air temperature | $K$ | ERA5 reanalysis | (Hersbach et al., 2020) | 2000-2002 |
| pr - Precipitation | $mm/day$ | GPCP version 2.3 | (Adler et al., 2016) | 2000-2002 |

$$E_{tot} = \frac{1}{P} \sum_{s=1}^{P} \frac{E_s}{E_{baseline,s}} \tag{2}$$

where P is the number of variables considered (P = 4, in the present work). The aggregated metric $E_{tot}$ represents a normalized distance between observations and a given model member, so that if $E_{tot} < 1$, the given member is considered to have a better skill than the baseline model version.

### 3.2 Surrogate Model

Surrogate models (or emulators) that are computationally inexpensive can be used to fully explore the ARPEGE parameter phase space at a limited cost. Here we train two different emulators to predict on one hand, the ARPEGE climatological skill (the mean state emulator thereafter), and the contrail radiative forcing (RF thereafter in the remainder of this section for clarity) on the other hand. The emulators are simply based on multi-linear regression and the predictors are the values of the 22 ARPEGE parameters that have been sampled in the PPE. The mean state emulator is built and trained to predict the five first PCs of the EOF analysis described in the previous section, expressed as follows:

$$Y = \sum_{j=1}^{K} a_j x_j + R \tag{3}$$

with $Y = PC_i$, the ith PC of the EOF analysis (from a total N = 5), $x_j$ the parameter value, $a_j$ the regression coefficient estimated based on the training of the model, R the intercept and K = 22 the number of perturbed parameters. The RF emulator is trained to predict directly the contrail radiative forcing values (in $mW.m^{-2}$, calculated as for Equation 3 with Y = RF). During the evaluation process, both emulators are trained on 150 members of the PPE, the 40 other members being used as the test dataset to estimate the out-of-sample error :

$$OSE = \sqrt{\sum_{i=1}^{40} \frac{(pred_i - true_i)^2}{40}} \tag{4}$$





### 3.3 Optimization Under Constraint

Our objective is to find the best-performing (based on the above total error score) model versions spanning a given range

of contrail radiative forcing, taken here as the range of contrail RF simulated by the PPE. To do so, optimal parameter values minimizing the total error $E_{tot}$ given by the mean state emulator described above have to be found for each bin of the discretized range of contrail RF simulated by the PPE. A constrained linear minimization optimizer (Virtanen et al., 2020) based on sequential least squares programming is used and the optimization process presented by Peatier et al. (2022) is followed step by step :

1. The emulators are used to produce a 100,000-members ensemble. A vector of discrete contrails radiative forcing is created, of length $n_{RF}$, such that $RF_i$ spans for all values of i the range of contrail RF seen in the ensemble in increments of 0.5 $mW.m^{-2}$.

   2. For each bin of contrail RF, the calibration which lies within the given bin (i.e., $RF_i - 5\,mW.m^{-2} < RF < RF_i + 5\,mW.m^{-2}$) and shows the lowest $E_{tot}$ based on the mean state emulator is selected. The corresponding parameter
values are then used to initialize the optimizer.

   3. The optimizer is then used to produce $\mathbf{P_1}$ - a 22 by $n_{RF}$ matrix with optimal parameter value estimates for each discrete bin of $RF_i$.

   4. $\mathbf{P_1}$ is a good first guess of optimal parameter values but is usually very noisy. We assume that parameters are varying smoothly across the contrail radiative forcing range (as suggested by (Neelin et al., 2010)) and repeatedly apply a
three-point moving average along this axis to each parameter value individually.

   5. These smoothed parameter values are then used to initialize another iteration of the optimizer, which produces a smoother final estimate of optimal calibrations $\mathbf{P_2}$.

   6. Finally, we sample a subset of 19 calibrations (the optimal ARPEGE candidates thereafter) from $\mathbf{P_2}$, spanning the whole contrail radiative forcing range, and use them to perform a new set of simulations with the full ARPEGE model.

### 3.4 Estimation of the Relative Likelihood Function

Assuming that the total optimal calibration errors follow a Gaussian distribution of width $\sigma$, the relative likelihood function of contrail radiative forcing, considering the parametric model $\theta = (E_{tot}(RF_i), \sigma^2)$, can be expressed as proposed in Peatier et al. (2022) :

$$\mathcal{L}(\theta, RF_i) \sim exp\left(-\frac{E_{tot}(RF_i)^2}{2.\sigma^2}\right) \tag{5}$$

with $E_{tot}(RF_i)$ the total error associated with the optimal calibration along the contrail radiative forcing range and $\sigma = OSE$. In addition, interpolation between errors for optimal ARPEGE candidates is used across the contrail radiative forcing range to compute the same estimate of the likelihood function.





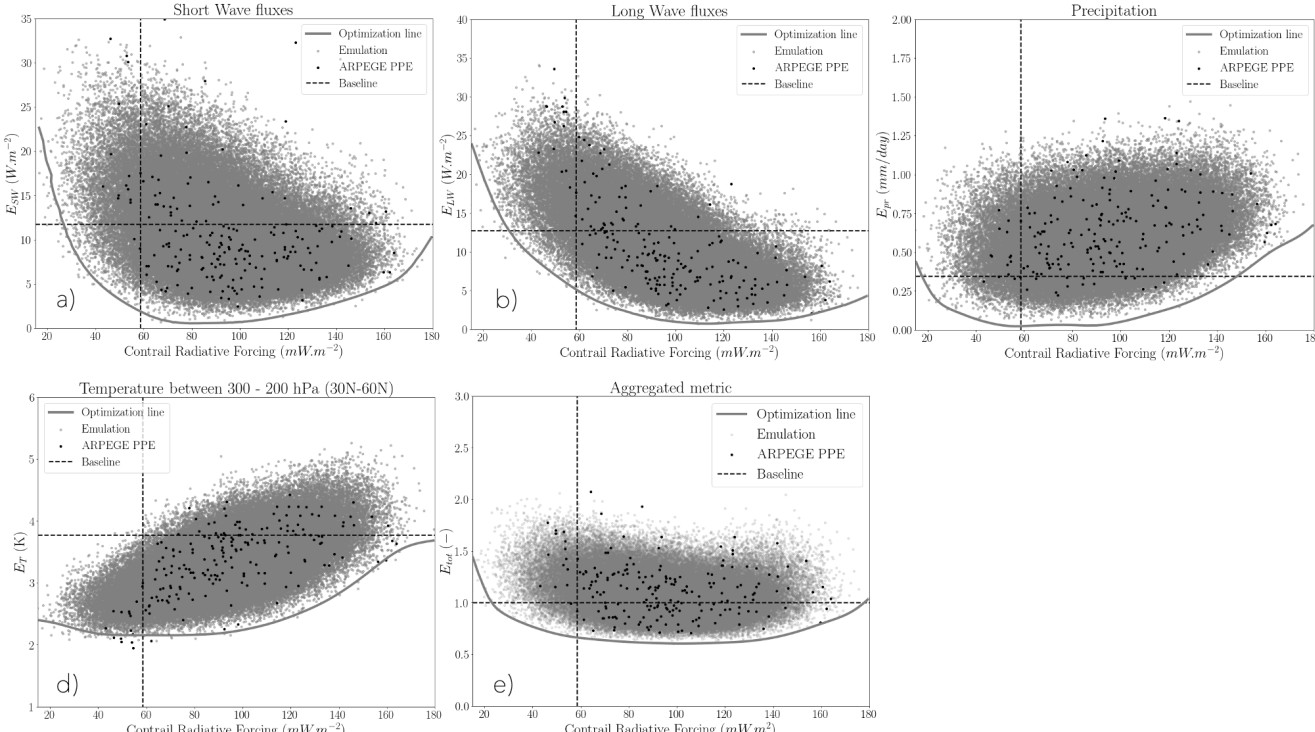

**Figure 6. Errors $E_s$ as a function of the contrail radiative forcing at the top of the atmosphere in the ARPEGE PPE (black dots) and the emulated ensemble (gray dots). The errors are computed for a) the short wave fluxes (SW) and (b) the long-wave fluxes (LW) at the top of the atmosphere, (c) the total precipitation, and (d) the temperature between 300-200hPa and 30N-60N. The four errors are then combined to compute the total error $E_{tot}$ as described in the performance assessment section. The baseline values of the contrail radiative forcing and errors are given by the dashed black lines. The solid gray lines correspond to optimized calibration with constraints on contrail radiative forcing (step 5 of the method) for the different metrics.**

## 4  Results

### 4.1  Calibration under constraints

Figure 6 shows the performance assessment of the ARPEGE ensemble and the emulated ensemble. The PPE gives a large range of contrail radiative forcing from 20 to 180 $mW.m^{-2}$. The reference value of the baseline simulation (57 $mW.m^{-2}$) lies on the right side of the distribution. Most of the PPE members predict air temperatures, short wave and long wave fluxes with better skill than the baseline version of the model. However, only a small number of PPE members have a better score than the reference for precipitation. In the end, based on the aggregated error metric, the PPE members are almost equally distributed on both sides of the baseline version error.

The emulators introduced in the previous section are used to produce an additional 100,000 members that can be compared with the baseline version. The emulated candidates span a slightly larger range of contrail radiative forcing than the PPE.



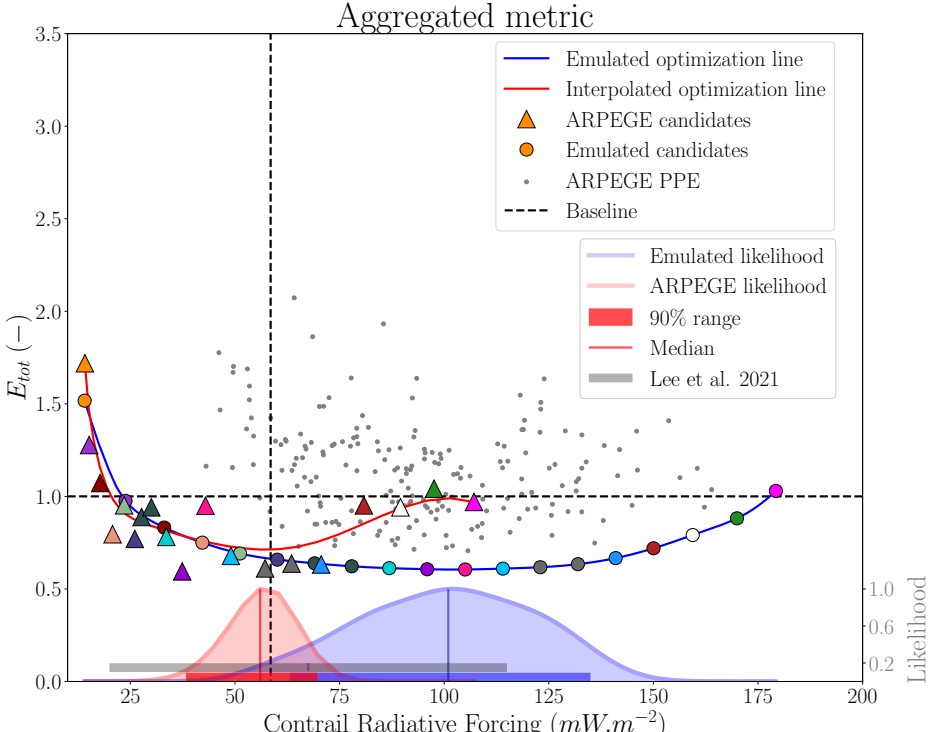

**Figure 7.** The total error $E_{tot}$ **as a function of the contrail radiative forcing at the top of the atmosphere for the selection of emulated optimized candidates (colored dots) and the ARPEGE-Climat simulations based on the same optimized calibrations (colored triangles). The two relative likelihoods of the contrail radiative forcing, estimated with Equation5, are represented by the shaded distributions, with the 5th to 95th percentiles range (90 % range) and the median in bright colors. The blue distribution is the initial likelihood function, estimated from the emulated optimal candidates (blue line). The red distribution is an adjusted likelihood function based on a smooth fit (using a cubic spline under tension interpolation) of ARPEGE optimal candidates error metric along the feedback range (red line). The 2006 estimates (90 % range and median) from Lee et al. (2021) are represented in gray.**

The performance of the emulation is rather similar to that of the PPE. However, the LW/SW metric scores are rather well distributed on both sides of the reference value which tends to shift the error score based on the aggregated metric to higher

values. The smooth optimization lines reveal an interesting behavior of the emulators: there are wide ranges of contrail radiative forcing with stable and minimum errors, especially for the aggregated metric (with a contrail radiative forcing range of 60-120 $mW.m^{-2}$).

This analysis indicates that it is possible to find several calibrations of ARPEGE that perform better (in the sense of the multivariate error metric) than the reference calibration. Furthermore, the magnitude of the contrail radiative forcing can be

multiplied by two from one *good* calibration to another. This last result, together with the contrail RF confidence interval for the year 2005 given above suggests that a substantial part of the uncertainty could be due to model differences related to parametric uncertainty.





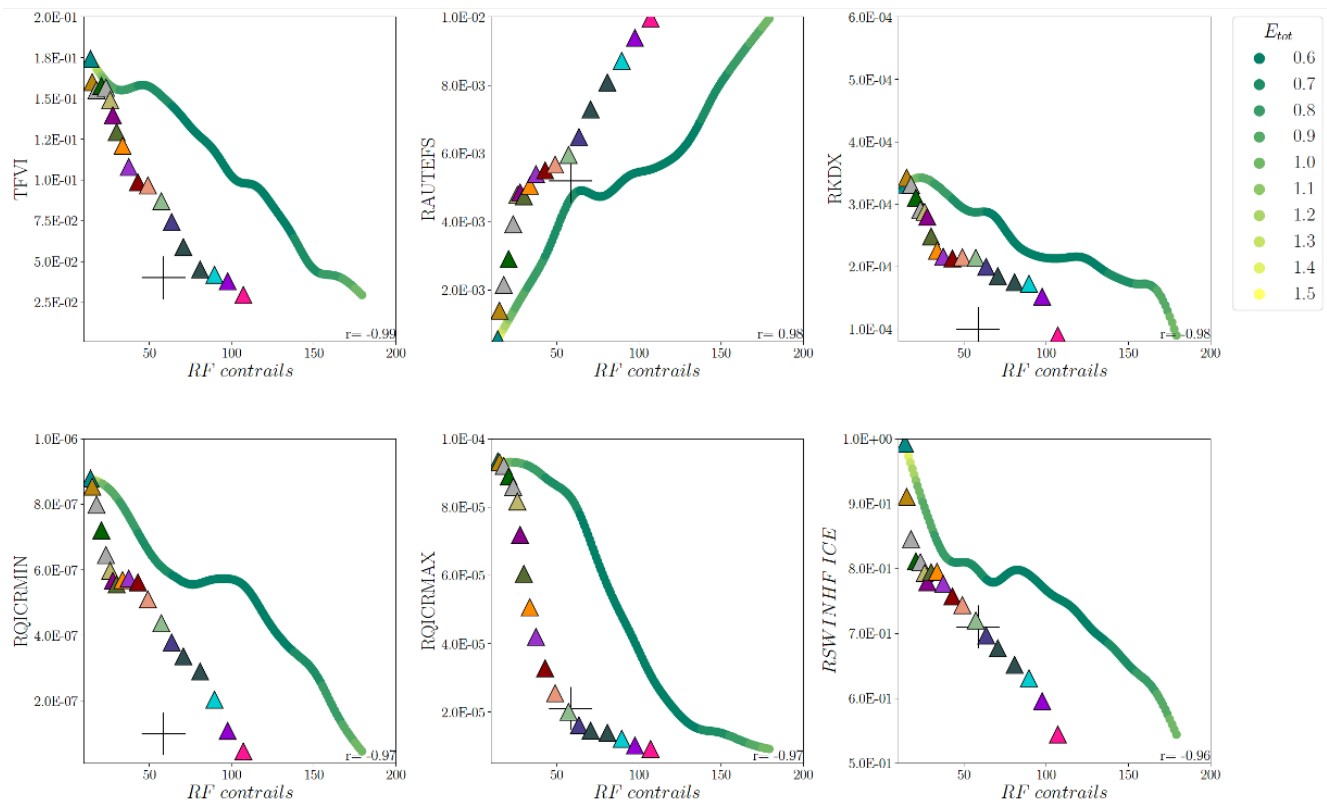

**Figure 8.** The values of the perturbed parameters that are the most correlated to the contrail radiative forcing according to the emulator. Correlation coefficients are given in the lower right part of each plot. The cross indicates the reference value of the parameter. The optimized calibrations from the emulator are represented by the dots and the associated total error $E_{tot}$ is represented by the color, from low values in green to high values in yellow. ARPEGE simulations using the 19 optimized candidates are represented by colored triangles. Description of the parameters by order of importance : TVFI is the falling speed of cloud ice crystals, RAUTEFS is the inverse timescale for ice autoconversion, RKDX is the maximum drag for the convective updraft vertical velocity, RQICRMIN (resp. RQICRMAX) is the critical ice content for ice autoconversion at low (resp. high) negative temperature, and RSWINHF_ICE is the ice cloud heterogeneity coefficient in the shortwave spectrum.

## 4.2    Toward an optimal calibration

Following the next steps of the method, 19 calibrations lying on the optimization line for the aggregated metric are selected to create a new ARPEGE ensemble to explore the range of contrail radiative forcing. Figure 7 represents this selection from the emulated ensemble and the performance of the corresponding ARPEGE simulations. The first observation, not explained yet, is the model trend to shift toward smaller constraint values (here the contrail radiative forcing) compared with the emulator. The second one is that 15 of the 19 simulations using the optimized calibrations perform better, given the aggregated metric, than the reference model. The scores are equal to or higher than those predicted by the emulator except for the eleventh candidate.



Finally, it is also interesting to notice that the prediction of contrail radiative forcing seems to be limited to a positive value (around 15 $mW.m^{-2}$) whether estimated by the PPE or the model.

The optimal calibrations from the PPE and the optimal ARPEGE candidates give very different estimates of the likelihood function. The range of contrail radiative forcing is halved between the first and the second estimate of the likelihood function. In the end, the median of the ARPEGE likelihood is 56 $mW.m^{-2}$, which is very close to Lee et al. (2021) best estimate.

The 5-95 % confidence interval (38-70$mW.m^{-2}$) is narrower and included in the one computed by Lee et al. (2021) (20-115 $mW.m^{-2}$).

We now assess the sensitivity of the contrail radiative forcing estimate to all the perturbed parameters in order to identify the most influential ones. We evaluate the relative importance of parameters using Pearson correlation coefficients between the parameters' values and the contrail radiative forcings. Figure 8 shows the six most important parameters according to the

sensitivity analysis performed with the emulator. The parameter with the highest correlation coefficient controls the fall speed of ice crystals. Three other parameters control the solid auto-conversion, more precisely its rate and the critical ice content for ice auto-conversion at low/high negative temperatures. This shows how important it is to accurately calibrate the contrails' lifetime to get a good estimate of the contrail radiative forcing. It is consistent with the results of Peatier (2022) who found that these parameters explain a large part of the PPE inter-member variance of high-cloud fraction. The last parameters that are

the most correlated to radiative forcing variations are the maximum drag for convective updraft vertical velocity and the ice cloud heterogeneity coefficients in the shortwave spectrum, meaning that convection and cloud optical properties also have an important contribution to contrail radiative forcing estimation.

## 5    Conclusions

An original contrail cirrus parameterization is presented in this paper. The cloud scheme from ARPEGE-Climat has been

adapted to allow supersaturation and an evolution law for contrail nebulosity, based on previous mesoscale simulations, is proposed. The representation of ice-supersaturated regions is in good agreement with observations. With the baseline calibration, the estimation of contrail radiative forcing for the year 2000 is 57.5 $mW.m^2$ which is also in good agreement with the last estimates in Lee et al. (2021).

Our second objective was to investigate the influence of model parametric uncertainty on the contrail radiative forcing es-

timation. A 200-member perturbed physics ensemble is generated and used to build statistical emulators to find optimized calibrations spanning a given range of contrail radiative forcing and having a skill similar or better than the reference atmospheric model. From all these calibrations, nineteen are kept to run additional simulations with ARPEGE-Climat. Fifteen of the ARPEGE-Climat candidates perform better than the reference according to the aggregated error metric. The resulting range of contrail radiative forcing from these ARPEGE-Climat simulations is halved compared with the emulator one. In light of our

results, a substantial part of uncertainties in the evaluation of the impact of contrails is parametric. Finally, a sensitivity analysis of the most influential parameters reveals that a few model parameters controlling contrails' lifetime are the most important for an accurate estimation of contrail radiative forcing in the ARPEGE-Climat model.



Future work will consider other relevant observational data to constrain even more the contrail radiative forcing. For instance, the relative humidity with respect to ice is a key control parameter for the formation of cirrus in the upper troposphere. An additional metric built on the relative humidity measurement from the IAGOS database would further strengthen our analysis. In addition, metrics based on cloud properties such as ice crystal effective radius and optical depth could also be introduced thanks to the MODIS data (https://modis.gsfc.nasa.gov/data/). Finally, the same optimization algorithm could also be applied to other atmospheric models with different contrail parameterizations to assess whether the key control atmospheric parameters are dependent on the contrail representation.

*Code availability.* Constrained optimization code is available for scientific research purposes from the authors upon request.

*Author contributions.* MP ran the calculation, investigated and produced the figures. SP provided code and guidance to the adapt her method to our study. MPM supported the setup of simulations. All authors participated in the interpretation of the results and the writing of the paper.

*Competing interests.* The authors declare that they have no conflict of interest.

*Acknowledgements.* The study was supported by the OCTAVIE project from the Occitanie (France) region (grant n°249586). This work benefited from previous work by Odile Thouron and David Saint-Martin on the adaptation of the cloud scheme for modeling condensation trails. The authors thank Romain Roehrig for discussions and support with the definition of the perturbed parameters and Dominique Bouniol for providing observational data on high clouds fraction.



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
