# Peer review of "Parametric Sensitivity and Constraint of Contrail Cirrus Radiative Forcing in the Atmospheric Component of CNRM-CM6-1"

_EGUsphere, 2023_

## Author Comment (AC1)

We would like to thank Dr Yun Qian, Atmospheric, Chemistry and Physics journal editor, for making this peer review process possible. We would like to thank the reviewers for the time spent on the paper, for the precious comments, suggestions, and the interesting questions asked. Those gave us the opportunity to clarify the context and the approach, to deepen the analysis and to support the conclusions. The quality of the paper has hopefully improved in the process. Following the reviewers' suggestions, a major revision of the paper has been conducted.

We have kept the two main parts of the draft (contrail parameterization – perturbed parameter ensemble). However, we have changed the structure of the paper to make it easier to understand our study. The new logic for this paper is the following: after introducing a new parameterization in the climate model (part 1), it is necessary to recalibrate the model to get a good level of performance (at least as good as the baseline). Here, we use a method (part 2) based on a perturbed parameter ensemble. The application of the method (part 3) gives us calibrations of ARPEGE-Climat, with the new ice-supersaturation/contrail parameterization, that are as good as the baseline version. These calibrations yield different values of the contrail radiative forcing, so the parametric uncertainties of using a climate model to estimate contrail radiative forcing are then analyzed (part 4). The introduction has been revised to provide a better insight into the context and our position relative to the existing literature on the topic, and to better state the objectives.

Our responses to the referee 1, referee 2 and community comments are described below. In the following, the comments from reviewer 1 are in purple, the comments from referee 2 are in green, the community comments are in orange and our answers are in black.

Referee 1:

This manuscript does two things. First it describes a contrail parameterization, and second it runs Perturbed Parameter Ensemble (PPE) with it to vary cloud parameters and see how they affect the contrail forcing. This paper is very confusing. It might be publishable with major revisions, or it might more usefully be two separate papers. Each part needs more description and detail as noted below.

Major comments;

1. The contrail parameterization is not described sufficiently, and there does not seem to be any uncertainty analysis of the different terms in it.

Following the reviewer suggestion, we have added some additional text to the section describing the contrail parameterization. An important point is that the ice supersaturation/contrail parameterization has to be consistent with the existing cloud scheme and its basic principles as shown in Figure 1 (see Ricard and Royer 1993 for a full description of the cloud scheme as well as Roehrig et al. 2020 to better understand how clouds are parameterized in the ARPEGE-Climat model, reference added P3 L90). Another

key point is the fact that contrail nebulosity is a prognostic variable that is added to natural cloud nebulosity. Some specific comments below also provide additional information.

The logic for this paper is the following: we know that the contrail parameterization is rather simple, yet it produces results that agree with other model estimates. Therefore, we think that it is more interesting to freeze the contrail parameterization, to include it in the atmospheric model and to investigate whether tuning ARPEGE adjustable parameters from other key parametrizations can lead to a good model performance (spatial pattern and magnitude of the effect) for the contrail radiative forcing, in addition to other classical performance metrics. This approach allows to run the contrail parametrization within the model component, to investigate the impact of this new parametrization on the model performance and to correct this impact through a calibration process. The second objective is then to study the influence of model parametric uncertainty due to the various adjustable parameters (from key physical parameterizations but that of the contrail) on the contrail radiative forcing. In the future we plan to develop a more complex contrail parameterization with more adjustable parameters and analyze its parametric sensitivity jointly with the full model physics package.

2. The simulations and methods for estimating radiative forcing are not described. How are the simulations run?

Following the reviewer suggestion, we have added more details on these aspects. Instantaneous radiative forcings are calculated from a single atmospheric model simulation with a double call to the radiative transfer code, one with and one without contrails (see P7 L158). If not specified, every value of radiative forcing presented in the paper is instantaneous radiative forcing (RF). "Radiative forcing" is used for the clarity in the text (see P1 L16).

3. The contrail parameterization simply does not seem physical. Please show how it relates to the Schmidt Appleman criteria. Given a temperature and humidity, what contrail fraction is predicted? It is impossible to assess the parameterization otherwise.

You can find in the Figure 1 below a representation of the saturation curves with respect to water (brown) and ice (yellow) as well as the mixing line (bleu) in a temperature-water vapor diagram. $G$ is the slope of the mixing line. $TLM$ is the threshold temperature given by the Schmidt-Appleman (SA) criteria. $TLC$ is the temperature below which the mixing line crosses the saturation curve (with respect to water). Our parameterization works for temperatures below $TLM$=235K (computed value using an efficiency of 0.4 and a pressure level of 300hPa). It is possible to introduce the probability $Psa$ (probability to satisfy SA criterion) to create a contrail between $TLM$ and TLC: $Psa = a/(a+b)$ (purple curve). You can find different values of $Psa$ in the Table 1. Even if it is possible to fit a second-order polynomial function to $Psa$, the Table shows a nearly constant value of 32% for each pressure level. Adding $Psa$ in our parameterization, simply by multiplying the right end side of Equation 1 in the main text, leads to a global contrail radiative forcing of 48 mW/m$^2$ (57 mW/m$^2$ for the baseline configuration of ARPEGE-Climat). Given our objectives and the other model uncertainties presented in the paper we believe our simple approach is fit-for-purpose. This has not been included in the text of the revised version of our paper. However, it could be added in the Supplement or in an Annex.

[Figure]

**Fig 1**. Representation of the saturation curves with respect to water (brown) and ice (yellow) as well as the mixing line (bleu) in a temperature-water vapor diagram. The different variables are discussed in the text.

| Efficiency | 0,308 | | | | | | |
|---|---|---|---|---|---|---|---|
| p (hPa) | G | G/p (1/K) | TLC (K) | TLM (K) | Prob (%) | TLM-TLC | formule 1 |
| | | | | | TLC<T<TLM | | |
| 200 | 1,35 | 0,00677 | 226,60 | 229,20 | 31,26% | 2,60 | 229,45 |
| 220 | 1,49 | 0,00677 | 223,75 | 230,25 | 33,00% | 6,50 # | 230,25 |
| 240 | 1,63 | 0,00677 | 224,50 | 231,25 | 32,80% | 6,75 # | 231,05 |
| 260 | 1,76 | 0,00677 | 225,25 | 232,00 | 32,67% | 6,75 # | 231,85 |
| 280 | 1,90 | 0,00677 | 226,00 | 232,75 | 32,50% | 6,75 # | 232,65 |
| 300 | 2,03 | 0,00677 | 226,75 | 233,50 | 32,40% | 6,75 # | 233,45 |
| 320 | 2,17 | 0,00677 | 227,50 | 234,25 | 32,27% | 6,75 # | 234,25 |
| 350 | 2,37 | 0,00677 | 228,20 | 235,00 | 32,30% | 6,80 # | 235,45 |
| 400 | 2,71 | 0,00677 | 229,80 | 236,60 | 32,12% | 6,80 # | 237,45 |

| Efficiency | 0,4 | | | | | | |
|---|---|---|---|---|---|---|---|
| p (hPa) | G | G/p (1/K) | TLC (K) | TLM (K) | Prob (%) | TLM-TLC | formule2 |
| | | | | | TLC<T<TLM | | |
| 200 | 1,56 | 0,00781 | 224,00 | 230,80 | 33,09% | 6,80 | 230,85 |
| 220 | 1,72 | 0,00781 | 224,75 | 231,75 | 32,70% | 7,00 # | 231,75 |
| 240 | 1,87 | 0,00781 | 225,80 | 232,60 | 32,71% | 6,80 # | 232,65 |
| 260 | 2,03 | 0,00781 | 226,80 | 233,60 | 32,50% | 6,80 # | 233,55 |
| 280 | 2,19 | 0,00781 | 227,60 | 234,40 | 32,40% | 6,80 # | 234,45 |
| 300 | 2,34 | 0,00781 | 228,20 | 235,00 | 32,30% | 6,80 # | 235,35 |
| 320 | 2,50 | 0,00781 | 229,00 | 235,80 | 32,20% | 6,80 # | 236,25 |
| 350 | 2,73 | 0,00781 | 229,80 | 236,60 | 32,12% | 6,80 # | 237,6 |
| 400 | 3,12 | 0,00781 | 231,40 | 238,20 | 31,98% | 6,80 # | 239,85 |

| Efficiency | 0,308 | formule 1 | TLM=230.25+0.04 (p -220) | prop=32,6% |
|---|---|---|---|---|
| Efficiency | 0,4 | formule 2 | TLM=231,75+0.045 (p -220) | prop=32,5% |

**Table 1**. Schmidt-Appleman criterion for different efficiencies and pressure levels.

4. Numbers are thrown around in the abstract and introduction that are not comparable (e.g. different dates), this needs to be cleaned up.

Thank you, it has been corrected.

5. I'm not sure what your RF represents since you do not describe the simulations or the method to estimate radiative forcing. How is the radiative forcing estimated? Is it Effective Radiative Forcing (ERF) including an adjustment in the model or RF?

Please see the answer to comment 2). No adjustment is taken into account, the instantaneous radiative forcings are calculated from a single simulation with a double call to the radiative transfer code.

6. I am very confused regarding how the emulator is built. The use of EOFs is pretty novel and interesting, but that means it needs more description and examples: I cannot seem to understand what you are doing, or how the results are derived.

Thanks for the comment. We have largely revised and hopefully improved the sections regarding the construction of the emulator. We now describe all the methodological steps one by one. Here is the use of EOFs is simply to reduce the dimensionality of the analysis. Answers to the different comments below should also help to clarify the method and derivation of the results.

7. I am concerned that all you have tested is the sensitivity to the background state of the model and evolution of clouds. It's not clear to me that you have really looked at the uncertainty of the contrails themselves (e.g. P and Lt in equation 1). See point 3.

For clarity, here P is the probability for the grid mesh to be saturated with respect to ice (= surface of the grid mesh that is saturated in the existing cloud scheme $a*$). It is computed at every time step (by the cloud scheme) and it is not an adjustable parameter of Eq. 1) (main text) It has been added to the text.

As mentioned in the answer to comment 1, one of the main objectives here is to better understand parametric uncertainties in the (instantaneous) contrail radiative forcing estimation due to key adjustable model parameters. Given the low complexity of the contrail parameterization, we freeze its associated parameters when investigating parametric uncertainty. Therefore, we are only considering uncertainty rising from the tuning of other key adjustable parameters of the atmospheric model. For instance, the growth rate parameter (dl/dt), used in the contrail parametrization is directly inferred from meso-scale simulations from Paoli et al. 2017. In the future, we plan to improve the realism and complexity of the parameterization and then fully address the question of parametric uncertainty, including parameters from the future contrail parameterization.

8. Finally, a minor point:  The English grammar could use a read through from a native speaker if possible, there are a number of mistakes, particularly the use of plurals. I sympathize, English is a really horrible language for this.

Thank you, we have tried to improve the grammar throughout the paper.

Page 1, L10 Are you simulating an RF or an ERF? They can differ by a factor of 2….if there is adjustment, then it's an ERF I believe?

See answers to major comments 2) and 5). It is instantaneous radiative forcing, there is no adjustment.

Page 1, L10: you should note that this estimate is for the year 2000. It is confusing for the reader.

See answer to major comment 4). The revised text is page 1, line 10.

Page 1, L17: You should state the 2000 or 2005 estimate from Lee et al 2021, otherwise the reader thinks you have a vast difference (also need to note year 2000 in abstract).

Thank you. In the introduction, we now give the more recent estimate for year 2018. The estimate and the confidence interval are there to illustrate the great uncertainty. Later on in the paper, we compare the result with Lee at al. 2021 best estimate for the year 2005. Dates have been added in the text when missing (abstract, P13 L254, P16 L290/L292).

Page 3, L68: Is cloud fraction in the scheme prognostic? I.e. da/dt will increase it?

Yes, cloud fraction is prognostic in the scheme as well as $x$ (= "are there already ice crystals or not?") to be consistent with existing prognostic variables (wind velocities, temperature, specific humidity, kinetic energy and the four hydrometeors). The sentence "the fraction of the grid mesh covered by contrails, $a$, is computed according to the following equation" (P3 L78) has been replaced in the new version by: "We introduce a prognostic variable $ac$, the fraction of the grid mesh covered by contrails, computed according to the following equation". Note that there was an error in the naming of the contrail cloud fraction. It is $ac$, not $a$ (which is the natural cirrus cloud fraction computed by the cloud scheme which is also prognostic).

Page 3, L69: is Qsat for liquid or ice? Or both?

The new scheme is activated for temperatures below 235K. For such temperatures, Qsat is with respect to ice. See the new version, page 3, line 93.

Page 3, L70: Grammar. "If there are no ice nuclei"

Thank you, corrected.

Page 3, L70: does this mean activated ice nuclei?

No, it is just "ice crystals", it has been corrected. See page 3, lines 93-94-95-97)

Page 3, L70: Suggest ice nuclei not be x since I think that used in equation 1 as distance, and it is confusing

Since the Figure belongs to David Saint-Martin (personal communication), we rather changed the distance in Eq. 1) (main text) from "dx" to "dy". See also P5 L110.

Page 3, L74: I don't understand what is happening. What is the difference between a and a*? The statement "in this case the contrail cirrus parameterization is activated whereas in every other case..." Is a = contrail coverage and a* = all clouds?

See answer to the above comment regarding the error in the naming of contrail coverage. $a*$ is the cloud coverage in the "existing cloud scheme" ($a*$ > 0 as soon as the air is ice-supersaturated with respect to ice). $a$ is the cloud coverage in the new cloud scheme ($a$ > 0 if there are no ice crystals in the grid box and the air is strongly supersaturated or if there already are ice crystals). $ac$ is the contrail coverage. $ac$ > 0 if case B) (see Fig. 1 in the paper) and follows equation 1). See equation 1) and page 4, line 102.

Page 3, L79: Can you illustrate what the contrail coverage tendency is as a function of say temperature and humidity? How does that compare to the Schmidt Appleman criteria?

Please see answer to major comment 3).

Page 3, L81: Saturated with respect to what? Ice?

For this temperature range, it is with respect to ice. See page 3, line 93.

Page 5, L93, Figure 2: Where is the Southern Hemisphere in the figure (right or left, use -lat for axis labels)? Also: please put some uncertainty bars on this based on internal variability. How long are the runs? Are these statistically different? How do you get different cloud fraction (singular cloud fraction is better English usage), in both hemispheres? Where are the contrails? Or is that embedded in the difference? Please explain. You are adding 2 things at once.

Labels used for the Southern Hemisphere are "90S", "60S" and "30S" for "ninety degrees south", "sixty degrees south" and "thirty degrees south" respectively. Interannual variability, represented as standard deviation, has been added to Figure 2.

For the atmospheric model and contrail parameterization validation of the model, we have performed 10-year long simulations. The annual mean value of contrail radiative forcing is 57.5 mW/m$^2$ $\pm$ 1.06 mW/m$^2$ (one standard deviation). For the PPE, we have run 3-year long simulations to reduce computing costs. According to a previous study (Peatier et al. 2021), 3-year long simulations were sufficient to get a robust estimate of the net global radiative budget.

Figure 2 shows the zonal mean of annual high-cloud fraction (corrected in Figure 2 caption) for CALIPSO data (black curve), the ten-year simulation using the "existing cloud scheme" (no supersaturation with respect to ice is allowed) and the ten-year simulation using the new cloud scheme/parameterization (allowing ice supersaturation and computing additional contrail coverage). Since the new cloud scheme allows supersaturation, a lower cloud cover could be expected. Figure 2 is presented to ensure that cloud climatology, especially at high altitudes, is still reasonably well simulated using the new cloud scheme.

Page 5, Fig 3: Great to see this. Can you add zonal mean latitude height panels of the ice supersaturation frequency from the model and AIRS.

We contacted the authors of Lamquin et al. 2012 to get access to the data. Unfortunately, the only AIRS data available are global annual means of the frequency of occurrence of ice supersaturation (ISS) for pressure levels 100-150, 150-200, 200-250, 250-300, 300-400, and 400-500 hPa. It is not relevant to plot the annual and zonal means (latitude-height panel) as suggested. Figure 2 shows the annual and zonal means of ISS occurrence frequency for the model. The ISS occurrence frequency is underestimated compared to Lamquin et al. 2012, especially in the tropics. Note that the large cold bias (4K) for pressure levels below 400 hPa may be at the origin of such discrepancies. In this paper, we focus on the layer where the maximum flight distance occurs (200-250 hPa). Figure 2 can be added in the supplement.

[Figure]

**Fig 2.** Annual and zonal mean of ice supersaturation occurrence frequency for ARPEGE.

Page 5, L105: again, show lat-height in Figure 3 as well.

Same answer as the previous comment.

Page 7, L117: what is the definition of contrails? Is this just da/dt per timestep from the parameterization? It's not contrail cirrus since it exactly matches the emissions locations.

Yes, contrail nebulosity follows Eq. 1) (main text). As mentioned in a previous comment, ac is a prognostic variable (see page 3, line 88) and ac is added as a new nebulosity to a (the natural cloud nebulosity) for the second radiative transfer call. ac is advected, as well as natural cloud nebulosity so contrail cirrus nebulosity does not exactly match the emissions locations, except for the initialization (see P5 L113)

Page 7, L117: maximum contrail coverage (singular I think is appropriate here, no of).

Thanks, corrected.

Page 7, L121: how is the net radiative effect calculated? Is it a difference between two runs? Or is it an off line calculation. You need to describe this more completely. Is this just a single number? What are the optical properties of a contrail? There needs to be much more detail here.

As previously mentioned, the (instantaneous) radiative forcing is computed by two radiative transfer code calls. Contrail optical properties are the same as natural cirrus. It is also a limitation of our method for the moment but efforts will be made to differentiate them in terms of crystal size and concentration (see page 5, line 115).

Page 7, L127: You should be clear that it is not the contrails that improve the clouds, it is the ice supersaturation used for cirrus clouds. Maybe you can do a run without contrails but with the new parameterization? Or have you done that in figure 5 already? Please clarify.

Thanks, our original sentence was misleading. In the introduction, we mention a contrail cirrus parameterization which underestimated by one order of magnitude the contrail radiative forcing (when "k" was just an adjustable parameter of the parameterization and not evolving with the temperature).  References to this parameterization have been removed as irrelevant.  The improvement referred to this parameterization.

That being said, the objective of the supersaturation parameterization was not to improve global cloud cover prediction even if, for example in the Antarctic, the parameterization helps to reduce the positive bias of total cloud cover as discussed in Roehrig et al. 2020. In the new version, the last paragraph of this section (P7 L127) has been changed to "To summarize, thanks to the ice-supersaturation/contrail cirrus parameterization in ARPEGE we are now able to estimate the contrail radiative forcing for the year 2000. Our estimate is in line with the most recent estimate from the literature. Comparison with cloud observation data (COSP) showed that high cloud climatology is still well simulated after the introduction of cloud-free and ice-supersaturated regions in the model."

Page 8, L145: how do five modes yield 3 variance values? And shouldn't all the EOFs add to 100%? Also, are these 2D (lat Lon) or 3D (lat, Lon, time) fields? I.e. radiative fluxes and precipitation are for each level or just surface (precip) and top of atmosphere (rad fluxes) as is more common?

Here the variance values represent the cumulative variance of the first five modes for the four variables (precipitation, radiative fluxes and temperature). As the cumulative variance is identical for the two radiative fluxes, there are only 3 values. All EOFs explain 100%, but please note that we are using only the first 5 modes in our analysis (variability is explained in descending order of eigenvectors). The temporal dimension has to be removed in order to perform the EOF analysis across the PPE, therefore the simulations have been averaged over the whole period. Precipitation and radiative fluxes are two-dimensional (latitude, longitude) fields. Temperature is initially a three-dimensional (height, latitude, longitude) field but here

it is pressure-averaged between 200 and 300hPa and ends up being two-dimensional (latitude, longitude). This is for information; it has not been added to the text.

Page 8, L146: I don't really follow this, since the method of using EOFs is not that common. Can you show an detailed example to help explain this for the reader?

We have completed the text (P10 L190-195) to provide more information to the reader and better explain the EOF method: "For each variable and ensemble member, the simulation is averaged over the whole period and the PPE mean is removed. The empirical orthogonal functions (EOFs) are then calculated to determine the dominant modes of variability across these 2D fields (lat, lon) of the PPE members. Contrary to the conventional EOF method, the temporal dimension is replaced by the ensemble dimension itself (Sanderson et al., 2008; Sexton et al., 2012). The resulting EOFs are spatial patterns, characterizing the variability of the ensemble variance, while their principal components (PCs) are the expansion coefficients showing the projection of each ensemble member onto the respective EOF."

L195 : EOFs are introduced here because they are used in the next section to build the surrogate models of the different fields (temperature, radiative fluxes and precipitation). Indeed, EOF analysis is often used to reduce the dimensionality of the problem producing a basis of vectors (modes) from which the fields can be expressed as linear combinations. Figure 2 (not added in the text) provides an example of the EOF analysis for surface temperature. Note that the surface temperature is not used to assess model performance in the paper, where we use temperature in the Northern Hemisphere upper troposphere mid-latitudes (30N –60N). Therefore, this Figure is only an illustration.

[Figure]

**Fig 3**. Illustration of the EOF analysis for surface temperature. In the top panel (a) the EOF analysis is performed within the 2D anomalies of ensemble members, the 3 first modes of variabilities are presented, as well as their associated PCs. The lower panel (b) presents an example of reconstruction of the 2D field of any PPE member using the EOFs and the PCs. In our study, the EOF basis is truncated after the fifth mode : q=5 in equation (b).

Page 9, L160: I thought the PCs were projections of ensemble members or something? How many EOFs are being used? Are they fixed? Again, I'm not following what is going on here. What does predicting the first 5 PCs get you?

Indeed, PCs are the projection of ensemble members averaged over time (2D field, lat x lon) onto the PPE EOFs basis truncated after the fifth mode of variability. The PPE EOF basis is fixed and with 5 PCs, associated with the 5 EOF modes, we can reconstruct the 2D field (lat x lon) of the simulations (see Fig. 2). Because the basis has been truncated, we do not have perfect reconstructions of the fields, but projecting the observations onto the EOF basis allows to compare the reconstructed fields of the simulations with the reconstructed fields of the observations through spatial RMSEs. Figure 3 presents the correlation between simple spatial RMSEs and EOF-based RMSEs for different variables of Peatier et al. (2022). Even though the amplitude of the error is under-estimated, the correlation of the errors across the PPE is very well captured (Fig. 3), validating the use of such metric for tuning purpose.

[Figure]

**Fig 4**. Correlation between a simple spatial RMSE (x axis) and the EOF-based RMSE (y axis) for each of the individual variable : (a) short wave fluxes ($E_{SW}$) , (b) long wave fluxes ($E_{LW}$), (c) surface temperature ($E_{tas}$) and (d) precipitation ($E_{pr}$). Figure from Peatier et al. (2022).

By using metric based on the EOF analysis, we can train statistical models to predict the PCs of ensemble members, which reduces the dimensionality of the prediction problem and increases the skill of the predictions. Indeed, it would be impossible to predict the >30000 grid points from the 30 perturbed parameters with the limited size of the training dataset available. By using the EOF analysis, we achieve acceptable predictions of the model performances.

Page 9, L164: Are you predicting the value at each location or the global average? LW, SW or total?

Unlike the other variables, for which we predict the PCs of the 2D fields, we predict directly the global average of the contrail net radiative forcing, as mentioned in the text. This makes it possible to use the prediction as a constraint in the optimization problem and to target specific values of contrail radiative forcing when identifying model candidates.

Page 10, L177: are you predicting the global mean value? Does this assume there is a unique set of parameters to get any contrail RF? What if the same RF is produced by a different set of parameters?

Yes, the global mean value is predicted. There is not necessarily a unique set of parameters for a given contrail radiative forcing. If the same radiative forcing is produced by a different

set of parameters, the two configurations are likely to be different regarding their performance on other climate metrics.

Page 10, L181: Again, I'm not following here. You have different parameters for each RFi bin? But if you have the lowest Etot doesn't that just specify the parameters and a given contrail RF for those parameters?

The Figure 4 below illustrates the different steps of the method. First, for each bin of contrail radiative forcing, the emulated calibrations (grey dots) with the minimum error are found (shaded red triangles on the figure). The fact that calibrations are chosen among the 100 000 emulated samples (not among the PPE) has been added in the new version, page 12 line 223. Then, these values are then used to initialize the optimizer (step 3 of the method). The result of the optimizer is given by the shaded red line. As it is written in step 4, this set of calibrations is very noisy and we try to smooth it as suggested by Neelin et al. 2010. The smoothed parameters (red bright triangles) are then used to initialize another iteration producing the smoothed optimized candidates (red line). Along the red optimization line, we select 19 calibrations (orange circles) to run with ARPEGE-Climat.

[Figure]

**Fig 5**. Representation of the different steps of the method for the aggregated metric. The PPE members are represented using black dots. The error and the contrail radiative forcing are given by the black dashed lines. Other symbols are discussed in the text.

The Figure 4 and an extended caption could be added in the Supplement.

 I don't understand how the optimized calibration can basically be out of sample of all the points. I obviously do not understand what you are doing.

See also the previous comment. The results from the emulator (grey dots), not the PPE values (black dots), with the minimum error for each bin of contrail radiative forcing are used to initialize the optimizer. Next, a smoothing is done. If you look at Figure 6 you can observe that the optimized line is close to the grey dots clouds.

Page 11, L203: Temperature yes, but the radiative fluxes appear near the middle of the the distributions (vertical axis of figure 6 right?).

When we look at Figure 6, we can see there are more black dots (PPE members) below the horizontal dashed black line than above (for temperature, SW and LW fluxes) which means these members perform better than the reference.

Page 12, L215: I do not follow this multiplying by two. Where did that come from?

If we consider a tolerance around the minimum total error value (pink circle in Figure 7), corresponding to a radiative forcing ~100mW/m$^2$, we can say that at least four configurations after and before the pink circle are also "good" from our metric point of view. These configurations correspond to contrail radiative forcings values from 70 to 130 so almost a factor of two.

Modification P14 L265: "Furthermore, the likelihood calculated from the emulation of optimized candidates (Figure 7, blue distribution) highlights that plausible models can present a large range of contrail radiative forcing, from 70 to 130 W/m2."

Page 12, L217: again, I don't follow this conclusion.

The fact that two "good" model configurations can lead to contrail radiative forcings differing by a factor 2 highlights the sensitivity of this variable to the model parameters. Our results confirm that the parametric uncertainty associated with the prediction of contrail radiative forcing is substantial in global circulation model.

Page 13, L222: the model tends to shift towards…

Corrected.

Page 13, L224: But a higher error score is bad right? Or good? I was reading from Figure 6 it was a minimization problem.

Yes, most of the optimized candidates perform better than the reference. The comment is here to underline the fact that the emulator underestimates the performance of the calibrations regarding our metric.

Page 14, L225: But this also means the emulator is not very effective at reproducing the results of the candidates, particularly with respect to contrail RF.

The emulated candidates are not represented in Figure 7, there only are the PPE members (grey dots). This can explain the "gap" between grey dots and circles on the left part of the figure. Circles are the optimized candidates (step 6 of the method) and triangles (with corresponding color) are ARPEGE runs using the optimized configurations.

Page 14, L234: note which years are being assessed.

The year has been added.

Page 14, L242: what about varying the input assumptions to the contrail parameterization? You have tested the sensitivity of overall clouds and contrails to parameters, which is interesting, but your contrail parameterization is very crude and not especially physical, what about testing the parameters in that?

See answers to previous comments. We argue that the fact that we use a simple parameterization (with only one adjustable parameter, the growth rate) makes a sensitivity study to its own single parameter a rather meaningless exercise. We argue that it is much more interesting to identify key parameters from other parameterizations that have an influence on the contrail radiative forcing (even though it is estimated based on a simple approach; yet the results from this simple parameterization seem to agree quite well with the best estimate from the current literature)

Page 14, L255: but you have just tested part of the uncertainty, you have not varied the contrail parameterization. You have just tested its sensitivity to the overall environmental state.

See answer to the above comment

Page 15, L264: I think you need to do more analysis of the contrail parameterization uncertainty.

Indeed, we plan in future work to improve our contrail parameterization based on a more complex representation of contrail cirrus with additional parameters. A complete uncertainty analysis would then be an important aspect of the parameterization evaluation process.

Referee 2:

The stated aims of this manuscript is to: (i) describe a new ice-supersaturation/contrail parameterisation in the ARPEGE model; (ii) evaluate the model parameterization against observations; and (iii) explore the parametric dependence of contrail radiative forcing to different calibrations of ARPEGE-Climat. Upon reading the manuscript, I would recommend the manuscript to be rejected on the following grounds:

The literature review in the Introduction is incomplete. For example, it does not provide a review of the contrail models that are currently available. It does not outline, describe, and review the main sources of uncertainty influencing the contrail climate forcing. Most importantly, it also does not provide a clear motivation to perform the stated research objectives. What is the novelty of this research, and how does it address the existing research gap?

Following the reviewer comments, we have substantially modified the main text. First, references such as Bier and Burkhardt 2022 and Gettelman et al. 2022 have been added in the introduction. To our knowledge, there seem to be only two GCMs able to estimate contrail radiative forcing: ECHAM and CESM2. As a matter of fact, these two estimates are used in Lee et al. 2021 review to propose a contrail radiative forcing best estimate of 67mW/m2 for the year 2005. This has also been added in the introduction.

We have also largely revised the description of the motivation for our research study. It is known that parametric uncertainty is a major source of uncertainty and is often not accounted for in climate assessments. The main objective of our study is to give an estimate of the parametric uncertainty regarding the magnitude of contrail radiative forcing. To do so, we use the atmospheric component, ARPEGE, of the CNRM-CM6-1 coupled climate model with a simple contrail parameterization (we note that this simple parameterization does provide contrail radiative forcing estimates that are within the contrail RF published range). At this stage, we argue that it is more important as a first step to look at the contrail RF parametric uncertainty coming from other physical parameterizations rather than exploring the uncertainty linked to the simple contrail parameterization parameters for which there is little if any expert knowledge. However, we do agree that it is important to develop more complex contrail parameterizations with additional variables and parameters and explore their parametric uncertainty. It is a line of research that we intend to pursue in the future.

After introducing a new parameterization in a climate model, there is a need to recalibrate the model to ensure a good performance at least as good as the original one, without the parameterization. One way to do it is based on expert knowledge and consists of the adjustment by hand of some adjustable parameters and verifying a posteriori that the new parameterization works at least as well as the previous one. Here, we use a perturbed parameter ensemble to do it in a more objective way. The method has been applied in other contexts (climate sensitivity for example) and to our knowledge, it is the first time that it is applied to explore and estimate the parametric uncertainty related to contrail radiative forcing. Our results suggest that there is a large variability among contrail radiative forcing estimates from one "plausible" model calibration to another. This is an important result to keep in mind when one wants to give a full account of the different sources of uncertainty regarding non-CO2 effects.

The proposed contrail cirrus parameterisation, i.e., Eq. (1), is very difficult to understand. The derivation, assumptions, and limitations of Eq. (1) cannot be found in the manuscript, thereby hindering the review process.

In the new version, we have tried to provide a more detailed description of the parameterization. Below are the answers to major comments 3 and 7 from the first reviewer regarding the limitations:

You can find in the Figure 1 below a representation of the saturation curves with respect to water (brown) and ice (yellow) as well as the mixing line (bleu) in a temperature-water vapor diagram. *G* is the slope of the mixing line. *TLM* is the threshold temperature given by the Schmidt-Appleman (SA) criteria. *TLC* is the temperature below which the mixing line crosses the saturation curve (with respect to water). Our parameterization works for temperatures below *TLM*=235K (computed value using an efficiency of 0.4 and a pressure level of 300hPa). It is possible to introduce the probability *Psa* (probability to satisfy SA criterion) to create a contrail between *TLM* and TLC: *Psa = a/(a+b)* (purple curve). You can find different values of *Psa* in the Table 1. Even if it is possible to fit a second-order polynomial function to *Psa*, the Table shows a nearly constant value of 32% for each pressure level. Adding *Psa* in our parameterization, simply by multiplying the right end side of Equation 1 in the main text, leads to a global contrail radiative forcing of 48 mW/m$^2$ (57 mW/m$^2$ for the baseline configuration of ARPEGE-Climat). Given our objectives and the other model uncertainties presented in the paper we believe our simple approach is fit-for-purpose. This has not been included in the text of the revised version of our paper. However, it could be added in the Supplement or in an Annex.

[Figure]

**Fig 1**. Representation of the saturation curves with respect to water (brown) and ice (yellow) as well as the mixing line (bleu) in a temperature-water vapor diagram. The different variables are discussed in the text.

Efficiency 0,308

| p (hPa) | G | G/p (1/K) | TLC (K) | TLM (K) | Prob (%) TLC<T<TLM | TLM-TLC | | formule 1 |
|---|---|---|---|---|---|---|---|---|
| 200 | 1,35 | 0,00677 | 226,60 | 229,20 | 31,26% | 2,60 | | 229,45 |
| 220 | 1,49 | 0,00677 | 223,75 | 230,25 | 33,00% | 6,50 | # | 230,25 |
| 240 | 1,63 | 0,00677 | 224,50 | 231,25 | 32,80% | 6,75 | # | 231,05 |
| 260 | 1,76 | 0,00677 | 225,25 | 232,00 | 32,67% | 6,75 | # | 231,85 |
| 280 | 1,90 | 0,00677 | 226,00 | 232,75 | 32,50% | 6,75 | # | 232,65 |
| 300 | 2,03 | 0,00677 | 226,75 | 233,50 | 32,40% | 6,75 | # | 233,45 |
| 320 | 2,17 | 0,00677 | 227,50 | 234,25 | 32,27% | 6,75 | # | 234,25 |
| 350 | 2,37 | 0,00677 | 228,20 | 235,00 | 32,30% | 6,80 | # | 235,45 |
| 400 | 2,71 | 0,00677 | 229,80 | 236,60 | 32,12% | 6,80 | # | 237,45 |

Efficiency 0,4

| p (hPa) | G | G/p (1/K) | TLC (K) | TLM (K) | Prob (%) TLC<T<TLM | TLM-TLC | | formule2 |
|---|---|---|---|---|---|---|---|---|
| 200 | 1,56 | 0,00781 | 224,00 | 230,80 | 33,09% | 6,80 | | 230,85 |
| 220 | 1,72 | 0,00781 | 224,75 | 231,75 | 32,70% | 7,00 | # | 231,75 |
| 240 | 1,87 | 0,00781 | 225,80 | 232,60 | 32,71% | 6,80 | # | 232,65 |
| 260 | 2,03 | 0,00781 | 226,80 | 233,60 | 32,50% | 6,80 | # | 233,55 |
| 280 | 2,19 | 0,00781 | 227,60 | 234,40 | 32,40% | 6,80 | # | 234,45 |
| 300 | 2,34 | 0,00781 | 228,20 | 235,00 | 32,30% | 6,80 | # | 235,35 |
| 320 | 2,50 | 0,00781 | 229,00 | 235,80 | 32,20% | 6,80 | # | 236,25 |
| 350 | 2,73 | 0,00781 | 229,80 | 236,60 | 32,12% | 6,80 | # | 237,6 |
| 400 | 3,12 | 0,00781 | 231,40 | 238,20 | 31,98% | 6,80 | # | 239,85 |

| Efficiency | 0,308 formule 1 | TLM=230.25+0.04 (p -220) | prop=32,6% |
|---|---|---|---|
| Efficiency | 0,4 formule 2 | TLM=231,75+0.045 (p -220) | prop=32,5% |

**Table 1**. Schmidt-Appleman criterion for different efficiencies and pressure levels.

For clarity, P (Eq 1. In the paper) is the probability for the grid mesh to be saturated with respect to ice (= surface of the grid mesh that is saturated in the existing cloud scheme $a*$). It is computed at every time step (by the cloud scheme) and it is not an adjustable parameter of Eq. 1) (main text) It has been added to the text.

One of the main objectives here is to better understand parametric uncertainties in the (instantaneous) contrail radiative forcing estimation due to key adjustable model parameters. Given the low complexity of the contrail parameterization, we freeze its associated parameters when investigating parametric uncertainty. Therefore, we are only considering uncertainty rising from the tuning of other key adjustable parameters of the atmospheric model. For instance, the growth rate parameter (dl/dt), used in the contrail parametrization is directly inferred from meso-scale simulations from Paoli et al. 2017. In the future, we plan to improve the realism and complexity of the parameterization and then fully address the question of parametric uncertainty, including parameters from the future contrail parameterization.

There are major discrepancies between the model and observations (see Fig. 2, Fig. 3, and Fig. 4) which the authors brushed off as "reasonable agreement" (Line 101) and "better agreement" (Line 115). What does "reasonable" and "better" mean? Please quantify these statements with statistical metrics.

As suggested by the reviewer, and in addition to Figures 3 and 4, we have now added Table 2. with supersaturation frequency from observations (MOZAIC, AIRS) as well as climate models (ECHAM4 and ARPEGE). The values come from Burkhardt et al. 2008. One can notice large differences between the two observational datasets. Our parameterization provides reasonable results, "reasonable" here meaning with values often within the observed range and close to those from another climate model. However, in winter the frequency is slightly underestimated leading to a seasonal cycle maximum peaking is shifted in fall instead of winter (page 6, line 146).

| Data/Seasons | DJF | MAM | JJA | SON |
|---|---|---|---|---|
| ECHAM4, 230 hPa | 0.19 | 0.21 | 0.18 | 0.19 |
| ECHAM4, 275 hPa | 0.18 | 0.18 | 0.17 | 0.17 |
| MOZAIC, 230 hPa | 0.27 | 0.19 | 0.18 | 0.24 |
| MOZAIC, corrected | 0.26 | 0.18 | 0.17 | 0.23 |
| AIRS, 250-300 hPa | 0.22 | 0.31 | 0.22 | 0.17 |
| ARPEGE, 230 hPa | 0.17 | 0.19 | 0.20 | 0.21 |

**Table 2**. Seasonal ISSR frequency from MOZAIC, AIRS and ECHAM4 versus ARPEGE. ISSR frequency is computed for midlatitudes (30N-60N/95W-35E). Values from Burkhardt et al. 2008

The manuscript then proceeds to develop a surrogate model on the grounds of computational efficiency, and calibrates it using the ARPEGE parameters. Given the significant discrepancies between the ARPEGE model and observations, as mentioned in the previous point, the question arises concerning the practical utility of these obtained results.

As mentioned above, the key issue here is observational uncertainty. The reviewer was previously criticizing our use of the word "reasonable" but the use of the expression "significant discrepancies" raises the same problem when observational uncertainty is large as shown in the above table. To be precise, we do not *calibrate* an emulator using the ARPEGE parameters. Instead, we build a statistical model to speed up and simplify the calibration of ARPEGE after adding the contrail parameterization. We have tried to clarify the elements regarding the method and the model evaluation in the new version of the draft.

The authors compared the simulated precipitation from the model with observations, i.e., Table 1, which makes no logical sense. How does precipitation influence contrail formation and its associated properties?

As previously mentioned, our objective is to derive an estimate of parametric uncertainty regarding contrail radiative forcing. Our method is based on the use of the surrogate model in an optimization under constraint exercise. The surrogate shows that a large range of contrail RF exists with different sets of parameters but the associated model versions are not all plausible (here a plausible model version is a model version that has an error less or equal

to that of the reference version, the error being derived from a multivariate metric based on model-observation comparison for precipitation, temperature and radiative fluxes). The optimization step then looks for model versions that have both the smallest error for each binning interval of the contrail RF range provided by the surrogate model and an error that is less or equal to that of the reference model. We have tried to clarify these aspects in the new versions.

The quality of Figures 6 and 8 are both very poor and unreadable.

We tried to improve the quality of Figures 6 and 8.

While there are also numerous minor comments that merit consideration, it might be more judicious to prioritise addressing the major comments as outlined above before delving into these minor comments. In light of these substantial concerns, I suggest the paper be rejected outright.

Community 1:

I would like to comment on the analysis presented in figure 4: you have compared the RHI distributions of ARPEGE, 1°x1° with the RHI distribution of the IAGOS data. IAGOS data are very local measurements, since they are measured every 4s.  How do you justify this?  Do you think that these two distributions are comparable?

Thank you for the comment. We agree that the IAGOS data are very local measurements. The data need to be handled carefully if we want to compare climatology. For example, is it possible to aggregate data to determine an average relative humidity over a flight distance corresponding to the size of our grid? In the present case, we have compared normalized probability distributions of ISSR occurrence, so this does not seem to be a problem.